# TOKEN REDUCTION IN VISION TRANSFORMERS VIA DISCRETE WAVELET DECOMPOSITION

## ABSTRACT

Vision Transformers (ViTs) have achieved remarkable performance across various computer vision tasks, yet their high computational cost remains a significant limitation in many real-world scenarios. As noted in prior studies, decreasing the number of tokens processed by the attention layers of a ViT directly reduces the required operations. Building on this idea, and drawing inspiration from signal processing, we reinterpret the token embeddings of a ViT layer as a signal, which allows us to apply the Discrete Wavelet Transform (DWT) to separate low- and high-frequency components. Guided by this insight, we present Token REduction via WAvelet decomposition (TREWA), a token-pruning strategy built upon DWT. For each image in a batch, TREWA selects a pruning level by comparing that image's attention entropy with the batch one. It then applies the DWT to the token embeddings and forwards only the low-frequency coefficients (i.e., those capturing the image's main semantic structure) to the next attention layer, discarding 50–75% of the tokens. We evaluate TREWA on four benchmark datasets in both pre-trained and training from scratch settings, comparing it against state-of-the-art pruning methods. Our results show a superior trade-off between accuracy and computational efficiency, validating the effectiveness of our frequency-domain token pruning strategy for accelerating ViTs.

## 1 INTRODUCTION

Vision Transformers (ViTs) (Dosovitskiy et al., 2021) have rapidly gained popularity in computer vision due to their ability to learn effective global representations through self-attention. Unlike convolutional models, which operate locally, ViTs transform images into sequences of tokenized patches, similar to Natural Language Processing (NLP) (Raghu et al., 2021), achieving competitive or superior performance in classification, detection, and segmentation tasks (Hassija et al., 2025). However, this representational power comes at a high computational cost (Keles et al., 2023) because each patch becomes a token that traverses the entire depth of the model, with quadratic complexity with respect to the number of tokens. This can be inefficient, as many tokens are redundant or uninformative (Nauen et al., 2025). In fact, not all tokens contribute equally to the semantics of the image (Haurum et al., 2023), and at greater depths, only those most relevant to the task tend to be useful (Diko et al., 2024). This observation has stimulated growing interest in token pruning or merging techniques, which aim to reduce the number of tokens processed by the model while preserving its performance (Marchetti et al., 2025). However, existing methods address this challenge primarily through additional learnable modules or adaptive attention mechanisms that estimate the importance of tokens in each layer (Rao et al., 2021; Fayyaz et al., 2022). Although effective, these solutions increase model complexity and often introduce computational overhead or require an additional training phase, complicating their integration into inference pipelines.

In this work, we propose an approach called Token REduction via WAvelet decomposition (TREWA), an alternative approach to token pruning different from traditional mechanisms. The key idea of TREWA is to treat the token embedding sequence in an attention layer as a multidimensional signal, which allows us to use frequency domain analysis tools, such as the Discrete Wavelet Transform (DWT) (Alessio, 2015), to efficiently reduce the number of tokens propagated through the ViT (Zhang, 2019; Fuad, 2017). In particular, for each input image in an attention layer, we first compute the entropy of the [CLS] token's attention distribution over patch tokens. This measure reflects the degree of concentration or dispersion of attention and serves as a proxy for the

model's confidence in its internal representation. Based on the entropy value, we assign a pruning level: images with low entropy (i.e., more focused attention) undergo stronger pruning, while those with high entropy are left unchanged. We then apply the DWT to the token embeddings, treating them as multidimensional signals. This decomposition separates high-frequency components, which are typically associated with noise or less relevant local details, from low-frequency components, which, in ViTs, are typically associated with the main semantic structure and information of the image (Wang et al., 2025). Depending on the assigned pruning level, we retain only the appropriate low-frequency components (Chowdhury & Khatun, 2012; Tian et al., 2023). This selection is performed in an unsupervised manner, without introducing any learnable parameters. To evaluate TREWA, we conduct extensive experiments on four benchmark datasets and two ViT architectures, considering both pre-trained and from-scratch settings, and demonstrate that TREWA achieves a superior accuracy-efficiency trade-off compared to state-of-the-art pruning and merging methods.

The main contributions of this work are:

- We propose TREWA, a novel token pruning strategy for ViTs based on the DWT, which retains only the low-frequency components of tokens, preserving the core semantic information.
- The pruning is applied progressively across the depth of the network and is dynamically adjusted per image using the entropy of the [CLS] token's attention distribution, enabling more aggressive pruning when attention is highly concentrated.
- TREWA is deterministic and parameter-free because it does not introduce additional learnable components, nor does it require architectural modifications or retraining.

## 2 RELATED WORK

The computational cost of ViTs grows linearly with depth and quadratically with the number of tokens (Keles et al., 2023), prompting the development of various pruning and merging strategies to reduce tokens while maintaining accuracy (Haurum et al., 2023). Several pruning techniques address the problem in different ways. Among these, TopK (Haurum et al., 2023) selects tokens based on the attention of the [CLS] token. Similarly, DynamicViT (Rao et al., 2021) predicts importance using a binary mask, while ATS (Fayyaz et al., 2022) samples tokens based on their relevance. In contrast, TRAM (Marchetti et al., 2025) discards tokens using attention graphs, whereas EvolutionViT (Liu et al., 2025) eliminates redundant patches through an evolutionary algorithm. Finally, MADTP (Cao et al., 2024) integrates multimodal guidance with dynamic pruning.

Alternatively, token merging methods aggregate similar representations by replacing them with merged tokens. Among these, TokenFusion (Kim et al., 2024) and AdaViT (Meng et al., 2022) use similarity metrics or learned merging rules; ToMe (Bolya et al., 2023) applies a lightweight algorithm that progressively merges tokens without requiring retraining; PatchMerger (PM) (Renggli et al., 2022) introduces learnable parameters for "soft" mergers; and LTM (Wang & Yang, 2025) uses masks to combine tokens into the most informative weights. Hybrid approaches, such as EViT (Liang et al., 2022), combine pruning and merging by eliminating tokens with low CLS attention and merging redundant ones, while RanMerFormer (Wang et al., 2024a) combines pruning and merging via bipartite soft matching on key vectors.

An emerging line of research has begun to explore the integration of wavelet transforms (Alessio, 2015) into Transformers, to improve computational efficiency or enrich the multi-scale representation of images (Cao & Zhao, 2023; Wang et al., 2024b). For example, wavelet-based tokenizers have recently been proposed as an efficient alternative to patch embeddings in ViTs (Zhu & Soricut, 2024). Instead, Li et al. (2022) proposes an architecture that integrates the Discrete Wavelet Transform (DWT) within a sliding window Transformer, exploiting frequency decomposition to recover coherent structures from noisy data. Yang & Seo (2023) integrates DWT into the architectural components of the Transformer, replacing the traditional token mixer with a multi-level frequency decomposition. Wave-ViT (Yao et al., 2022) proposes integrating the DWT into ViTs as an invertible alternative to traditional downsampling on keys and values in attention.

Unlike pruning and merging methods that rely on learnable modules or static rules, TREWA is deterministic, interpretable, and adaptive: it leverages token attention and DWT-based frequency decomposition to prune tokens by retaining only low-frequency semantic components. Furthermore,

different from works that integrate DWT into ViTs for multiscale representations, we use it solely as a module during training and inference, without altering the ViT or adding parameters.

## 3 PRELIMINARIES

### 3.1 VISION TRANSFORMERS

A Vision Transformer (ViT) adapts the Transformer architecture to visual data. Although first introduced for image classification and later applied to many other tasks in computer vision, in this work, we employ a ViT solely as an image classifier. Let $I \in \mathbb{R}^{H \times W \times C}$ be an input image. The ViT divides $I$ into $N$ non-overlapping square patches of size $p \times p$ pixels, each of which is embedded into a vector space $\mathbb{R}^d$, generating a sequence of embedded tokens $X = \{x_1, \ldots, x_N\}$, preceded by a special token for classification [CLS]. The ViT model, denoted by $T$, consists of $L$ Transformer layers, each of which applies self-attention and feed-forward on all tokens. At the end of the $L$ layers, the [CLS] is used for image classification.

### 3.2 DISCRETE WAVELET TRANSFORM

The Discrete Wavelet Transform (DWT) decomposes a signal into components that are simultaneously localized in space (or time) and frequency. Unlike the Fourier Transform, which supplies only global frequency information, the DWT employs scaled and shifted wavelets to deliver a multiresolution view. Each decomposition stage filters the input with complementary low-pass and high-pass kernels, downsamples the outputs, and thus generates sub-bands: the low-frequency, or approximation, coefficients preserve the stable global structure, whereas the high-frequency, or detail, coefficients represent sharp changes or edges, and much of the noise (Li et al., 2021). Because this separation is both invertible and computationally efficient, DWT has become a useful tool for image compression, denoising, and broader multiscale analysis of visual, audio, and textual data. The DWT is composed of a low-pass filter $g$ and a high-pass filter $h$, both derived from the same mother wavelet $\psi$. After filtering, the signal is downsampled by a factor of two. For a one-dimensional input signal $x$, the approximation and detail coefficients are:

$$cA[i] = \sum_k x[k] \, g[2i - k], \qquad cD[i] = \sum_k x[k] \, h[2i - k]$$

The approximation coefficients $cA$ contain the low-frequency content that describes the global structure, whereas the detail coefficients $cD$ isolate the high-frequency content that captures local variations, fine details, and noise. Recursively applying the transform only to the approximation band $cA$ generates a depth $j$ decomposition $cA^{(j)}$ in which each successive level retains increasingly abstract semantic information.

## 4 METHODOLOGY

### 4.1 OVERVIEW

TREWA introduces a dynamic token pruning strategy for Vision Transformers, applied at selected layers during inference. For each image, we analyze the distribution of the attention weights from the classification token ([CLS]) to the other tokens, using Shannon entropy as a measure of dispersion. For every image in the batch, we first calculate the entropy of the [CLS]-token attention distribution across all tokens. A high-entropy (more uniform) distribution suggests that information is spread broadly, so we apply lighter pruning. Conversely, a low-entropy (more peaked) distribution indicates that attention is concentrated on a few tokens, allowing us to prune more aggressively. Afterwards, the token embeddings of an image are then decomposed via the DWT, which separates low- and high-frequency components. Only the low-frequency part, representing semantic information, is retained and propagated to deeper layers. The pruning intensity is controlled by the number of recursive DWT applications, which is determined adaptively based on the entropy computed for each image.

## 4.2 PRUNING STRATEGY

For clarity, we present the procedure on a single attention layer of a Vision Transformer, but the same steps can be applied to any number of layers.

Consider a generic self-attention layer $l \in L$. Let $\mathcal{B} = \{I_1, \ldots, I_B\}$, with $B > 1$, be a batch of input images. Each image $I_b$ is tokenized into $N$ patch tokens and one special classification token [CLS]:

$$X_b = \{ [CLS]_b, \, x_{b_1}, \, x_{b_2}, \, \ldots, \, x_{b_N} \} \in \mathbb{R}^{(N+1) \times d},$$

where $[CLS]_b$ denotes the classification token of image $I_b$ and $d$ is the embedding dimension. The remaining $N$ tokens $\{x_{b_1}, \ldots, x_{b_N}\}$ are the embeddings of the image patches. From the multi-head self-attention tensor $A_b \in \mathbb{R}^{H \times (N+1) \times (N+1)}$ computed at layer $l$ for image $I_b$, we first average over the $H$ attention heads. We then isolate the row corresponding to the $[CLS]_b$ token, discard its self-attention coefficient, and obtain the attention vector $a_b \in \mathbb{R}^N$, which contains the attention weights assigned by $[CLS]_b$ to the $N$ patch tokens only.

### 4.2.1 ENTROPY-BASED POLICY

To quantify how uniformly this attention is spread over the $N$ patch tokens, we measure its Shannon entropy:

$$h_b \; = \; - \sum_{i=1}^{N} a_b[i] \, \log a_b[i]$$

where $a_b[i]$ is the weight assigned by [CLS] to the $i$-th patch token of image $I_b$. A high entropy indicates that attention is broadly distributed, suggesting semantic complexity, whereas a low entropy reveals a focus on a few tokens and signals that more aggressive pruning may be tolerated (Zhang et al., 2025; Lee & Kim, 2024).For the current batch, we collect the entropies for each image $H = \{h_1, \ldots, h_B\}$, with $|H| > 1$, compute their mean $\mu_H$ and standard deviation $\sigma_H$, and define an adaptive threshold:

$$\tau_H \; = \; \mu_H - \sigma_H.$$

Each image is then assigned a discrete pruning level that controls how many tokens will be removed: images with $h_b \leq \tau_H$, whose attention is concentrated and thus easier to summarize, receive a stronger pruning level, whereas images with $h_b > \tau_H$ are preserved more thoroughly. In this way, we obtain a parameter-free rule that identifies the images that can safely undergo stronger pruning. Since both $\mu_H$ and $\sigma_H$ are recomputed for every batch, the threshold automatically follows variations in image complexity.

### 4.2.2 DISCRETE WAVELET TRANSFORM

For every image $I_b$ that requires pruning, we apply the 1D Discrete Wavelet Transform along the token sequence $\{x_{b1}, x_{b2}, \ldots, x_{bN}\}$. The intuition is to interpret the embedding sequence as a multidimensional signal; hence, the transform separates low-frequency content, which conveys the stable global semantics, from high-frequency content, which mostly carries local variations and noise. Formally, using a mother wavelet $\psi$ and a depth $j$ defined by the entropy-based policy:

$$\text{DWT}(X_b, \psi, j) \; = \; \left\{ cA_b^{(j)}, \; cD_b^{(j)} \right\}$$

where $cA_b^{(j)}$ holds the low-frequency approximation coefficients and $cD_b^{(j)}$ collects the high-frequency detail coefficients. The recursion depth is assigned adaptively:

$$j = \begin{cases} 1 & \text{if } h_b > \tau_H \\ 2 & \text{if } h_b \leq \tau_H \end{cases}$$

with $h_b$ the attention entropy of image $I_b$ and $\tau_H$ the adaptive threshold. Images whose attention is strongly concentrated ($h_b \leq \tau_H$) therefore undergo a more severe pruning. For $j = 1$ we keep only $cA_b^{(1)} \in \mathbb{R}^{\frac{N}{2} \times d}$. For $j = 2$ we apply a second transform to $cA_b^{(1)}$ and retain $cA_b^{(2)} \in \mathbb{R}^{\frac{N}{4} \times d}$. Afterwards, we replace the original token signal $X_b$ for the next attention layer with the pruned one computed by our approach:

$$X_b^{\text{new}} = \left\{ [CLS]_b \right\} \cup cA_b^{(j)}$$

Therefore, each image in a batch is effectively pruned by the factor $1/2^j$. Following this reasoning, our approach yields a batch-level token reduction between $50\%$ and $75\%$. Since the computational and memory complexity of self-attention scales quadratically with the sequence length, the subsequent attention block works with a significantly reduced cost.

## 5 EXPERIMENTS

### 5.1 EXPERIMENTAL SETUP

**Baselines** To evaluate the performance of TREWA, we compare our pruning strategy with six state-of-the-art methods: TRAM (Marchetti et al., 2025), ATS (Fayyaz et al., 2022), TopK (Haurum et al., 2023), PM (Renggli et al., 2022), ToMe (Bolya et al., 2023), and EViT (Liang et al., 2022). These methods include strategies based on pruning, merging, and token reordering, and constitute a broad spectrum of efficient approaches for ViT (Haurum et al., 2023). For a fair comparison, we set the pruning rate of all baseline methods to 50%, ensuring a consistent token reduction across approaches. Furthermore, we use $haar$ as the wavelet $\psi$ in TREWA.

**Models** The experiments are conducted on two ViT architectures: ViT-Base (ViT-B) and ViT-Small (ViT-S). ViT-B uses an embedding dimension of 768, 12 attention heads, an MLP dimension of 3072, and 12 layers, with a 16 × 16 patch size. ViT-S adopts an embedding dimension of 384, 6 heads, an MLP dimension of 1536, and 12 layers, with the same 16 × 16 patch size. In all approaches, we inserted the pruning modules at layers 5, 7, and 9, following a common configuration to ensure a fair comparison.

**Datasets** The evaluations were performed on four datasets commonly used for image classification: FashionMNIST (FMNIST) (Xiao et al., 2017), a grayscale dataset of 28×28 images of clothing classes; CIFAR10 (Krizhevsky et al., 2010), consisting of 32×32 color images across 10 object classes; Imagenette (Howard, 2020), a subset of ImageNet (Deng et al., 2009) with 224x224 color images from 10 classes; ImageNet-1k (Russakovsky et al., 2015), a subset of ImageNet consisting of 1.2 million training images of size 224x224 across 1,000 object classes, with a standard validation set of 50,000 images.

**Metrics** To compare the different approaches, we consider accuracy as the main measure of classification performance, Giga FLoating Point Operations (GFLOPs) to estimate computational cost, Frames Per Seconds (FPS) to capture inference speed. Reported accuracies are the mean ± variance over 10 runs with different random seeds to capture training stochasticity.

**Hardware Configuration** All experiments were performed on a machine with an Intel(R) Xeon(R) W5-3435X 3.10 GHz CPU, 128 GB of RAM, and an NVIDIA RTX A2000 GPU with 12 GB of memory.

### 5.2 QUANTITATIVE RESULTS

#### 5.2.1 PRE-TRAINED SETTING

In this section, we evaluate the performance of TREWA in a pre-trained setting. For ImageNet-1k, we used publicly available ViT-B and ViT-S models pre-trained on ImageNet, so no further adjustments were required. In contrast, FMNIST, CIFAR10, and Imagenette have no official pre-trained ViT models, so we fine-tuned the ImageNet pre-trained ViT-B and ViT-S weights on these datasets for 5 epochs, respectively. All input images are rescaled to $224 \times 224$ pixels, to match the

input resolution expected by the pre-trained models. The results are reported in Tables 1 and 2, with the best values in bold and the second-best values underlined.

Table 1: Performance of the evaluated approaches on ViT-B in the pre-trained setting.

| Model | FPS ↑ | GFLOPs ↓ | Accuracy ↑ | | | |
|---|---|---|---|---|---|---|
| | | | FMNIST | CIFAR10 | Imagenette | ImageNet-1k |
| ViT-B | 341 | 16.880 | 93.21±.07 | 95.72±.11 | 97.38±.15 | 80.32±.03 |
| TREWA | **538** | **9.218** | 92.79±.09 | **94.33**±.08 | **95.40**±.07 | **79.55**±.07 |
| TRAM | 500 | 11.849 | **93.05**±.10 | 93.68±.09 | 94.71±.08 | 79.44±.03 |
| ATS | 503 | 10.236 | 92.75±.08 | 92.96±.07 | 94.35±.10 | 78.96±.05 |
| PM | 528 | 11.496 | 91.01±.09 | 86.44±.13 | 87.09±.11 | - |
| TopK | 490 | 11.851 | 92.68±.07 | 93.19±.06 | 94.59±.08 | 77.71±.06 |
| ToMe | 493 | 11.878 | 92.42±.08 | 93.21±.09 | 93.70±.07 | 78.30±.04 |
| EViT | 495 | 11.932 | 92.20±.09 | 92.90±.08 | 94.26±.06 | 78.82±.09 |

Table 1 shows that, considering the ViT-B model, TREWA is the most efficient, outperforming PM by 1.89% in FPS and having 9.95% fewer GFLOPS than ATS. In terms of accuracy, TREWA achieves the best performance on the CIFAR10, Imagenette, and ImageNet-1k datasets, and ranks second on FMNIST after TRAM. We cannot report PM results on ImageNet-1k because PM requires training its own token-merging parameters, which is incompatible with the pre-trained setting, where the model is not further trained.

Table 2: Performance of the evaluated approaches on ViT-S in the pre-trained setting.

| Model | FPS ↑ | GFLOPs ↓ | Accuracy ↑ | | | |
|---|---|---|---|---|---|---|
| | | | FMNIST | CIFAR10 | Imagenette | ImageNet-1k |
| ViT-S | 853 | 4.257 | 93.73±.11 | 96.52±.05 | 98.12±.07 | 72.16±.04 |
| TREWA | 1,231 | **2.321** | **94.02**±.10 | **95.81**±.09 | **96.92**±.11 | **69.62**±.06 |
| TRAM | 1,018 | 2.996 | 93.58±.15 | 95.60±.13 | 95.49±.14 | 69.58±.07 |
| ATS | 1,016 | 2.610 | 93.39±.09 | 93.88±.10 | 95.24±.12 | 69.50±.08 |
| PM | **1,307** | 2.908 | 92.91±.08 | 88.01±.15 | 87.73±.12 | - |
| TopK | 1,186 | 2.997 | 93.20±.09 | 94.24±.08 | 95.45±.10 | 69.11±.03 |
| ToMe | 1,259 | 3.012 | 93.32±.07 | 95.19±.11 | 95.08±.13 | 69.60±.06 |
| EViT | 1,228 | 3.017 | 92.85±.10 | 94.97±.10 | 94.79±.09 | 69.19±.05 |

As for ViT-S, Table 2 shows that TREWA represents the best compromise between accuracy and computational efficiency. Although it is slightly slower than PM, with 5.81% fewer FPS, it is the most computationally efficient, with 11.07% lower GFLOPs consumption than ATS, which ranks as the second-best method. Finally, in terms of accuracy, TREWA also achieves the best performance on all datasets considered.

### 5.2.2 TRAINING FROM SCRATCH SETTING

In this section, we compare TREWA with state-of-the-art pruning and merging methods, training ViT models from scratch on selected datasets. The goal is to evaluate the effectiveness of pruning when applied throughout the entire learning process, without pre-trained weights. Specifically, all models were trained for 50 epochs on the Imagenette dataset and for 10 epochs on the CIFAR10 and FMNIST datasets. In addition, the input images were resized to $160 \times 160$ pixels to standardize the resolution across different experiments. Due to computational constraints, we do not report ImageNet-1k results, as training from scratch on millions of images is infeasible in our current setting. The results are shown in Tables 3 and 4.

Table 3 shows the results obtained by training ViT-B models from scratch. Our method is confirmed as the most efficient among those analyzed, being the fastest in terms of FPS and the least expensive in terms of GFLOPs. Compared to TopK, which is the second-best method in terms of speed, we obtain an increase of 8.22%, while compared to ATS, which is second in terms of computational efficiency, we reduce GFLOPs by 13.20%. In terms of accuracy, TREWA achieves the best results

on Imagenette and FMNIST datasets, while on CIFAR10 it is very close to TRAM, which is slightly superior.

Table 3: Performance of the evaluated approaches on ViT-B in the training-from-scratch setting.

| Model | FPS ↑ | GFLOPs ↓ | Accuracy ↑ | | |
|-------|-------|----------|------------|--|--|
| | | | FMNIST | CIFAR10 | Imagenette |
| ViT-B | 822 | 8.650 | 90.60±.16 | 68.17±.06 | 70.37±.11 |
| TREWA | **1,132** | **4.616** | **90.64**±.09 | 67.14±.08 | **69.48**±.10 |
| TRAM | 1,002 | 6.085 | 90.49±.08 | **67.88**±.11 | 69.07±.09 |
| ATS | 956 | 5.318 | 90.15±.07 | 66.61±.12 | 68.03±.10 |
| PM | 1,031 | 5.864 | 90.41±.10 | 65.29±.09 | 65.69±.11 |
| TopK | 1,046 | 6.086 | 90.53±.12 | 66.01±.07 | 68.89±.08 |
| ToMe | 1,011 | 6.185 | 89.84±.09 | 65.86±.10 | 68.35±.13 |
| EViT | 989 | 6.198 | 89.94±.10 | 64.96±.15 | 68.49±.06 |

Table 4 shows the results obtained when ViT-S models are trained from scratch. Our method stands out for its high computational efficiency, achieving the lowest GFLOPs value among all the methods analyzed. Compared to ATS, which is the second most efficient in this respect, we achieve a reduction of 13.33%. In terms of FPS, TREWA is second only to PM, with a margin of 2.64%. In terms of accuracy, we achieve the best results on CIFAR10 and Imagenette, while on FMNIST, we come close to the best result, with a minimal difference compared to TRAM.

Table 4: Performance of the evaluated approaches on ViT-S in the training-from-scratch setting.

| Model | FPS ↑ | GFLOPs ↓ | Accuracy ↑ | | |
|-------|-------|----------|------------|--|--|
| | | | FMNIST | CIFAR10 | Imagenette |
| ViT-S | 2,012 | 2.180 | 90.76±.05 | 66.76±.14 | 68.25±.11 |
| TREWA | 2,435 | **1.170** | 90.65±.07 | **66.37**±.10 | **67.04**±.11 |
| TRAM | 2,489 | 1.538 | **90.81**±.08 | 66.21±.09 | 66.73±.10 |
| ATS | 1,784 | 1.350 | 90.77±.09 | 66.29±.10 | 65.14±.12 |
| PM | **2,501** | 1.483 | 90.46±.08 | 65.70±.11 | 62.75±.13 |
| TopK | 2,452 | 1.538 | 90.01±.07 | 66.13±.09 | 66.65±.10 |
| ToMe | 2,397 | 1.558 | 89.48±.08 | 64.93±.10 | 65.39±.09 |
| EViT | 2,364 | 1.573 | 90.25±.07 | 62.36±.11 | 65.53±.10 |

## 5.3 ABLATION STUDY

In this section, we analyze the impact of the main design choices of TREWA through an ablation study. In particular, we evaluate three aspects: the pruning level assignment strategy, the type of spectral component retained after the DWT transformation, and the choice of the wavelet used in the decomposition. All experiments are conducted by training the ViT-B model from scratch on the FMNIST, CIFAR10, and Imagenette datasets, keeping the training conditions constant.

As a first variation, we analyze the method for dynamically choosing the pruning to be performed. The first approach ($\tau$) employs our adaptive threshold, defined as the batch-wise mean of the entropy values minus their standard deviation. The second method ($Quantile$) partitions the entropy distribution of the images within a batch into four quartiles. Images in the first quartile, which exhibit low entropy and therefore high confidence, are subjected to the most aggressive pruning ($j = 3$). Those in the second quartile undergo moderately strong pruning ($j = 2$), while images in the third and fourth quartiles, associated with higher uncertainty, are pruned more conservatively ($j = 1$), in accordance with the lower concentration of attention. The third method ($Gini$) considers the Gini Index (Farris, 2010) calculated on the attention distribution of the token [CLS]. If the value is less than 0.5, the DWT has $j = 1$; otherwise, $j = 2$ is applied. Unlike the previous methods, this approach bases the pruning decision on the inequality of attention distribution rather than entropy. The corresponding results are reported in Table 5.

From the analysis of Table 5, we note that the adaptive threshold $\tau$ represents the most effective choice for assigning the pruning level. While it yields a slightly lower frame rate compared to the

Table 5: Performance of the evaluated approaches when training ViT-B from scratch with different pruning thresholds.

| Method | FPS ↑ | GFLOPs ↓ | Accuracy ↑ | | |
|---|---|---|---|---|---|
| | | | FMNIST | CIFAR10 | Imagenette |
| $\tau$ | 1,132 | **4.616** | **90.64**$_{\pm.12}$ | **67.14**$_{\pm.08}$ | **69.48**$_{\pm.10}$ |
| $Quantile$ | 1,064 | **4.616** | 89.11$_{\pm.14}$ | 66.53$_{\pm.07}$ | 68.12$_{\pm.09}$ |
| $Gini$ | **1,221** | **4.616** | 89.72$_{\pm.09}$ | 67.04$_{\pm.07}$ | 68.76$_{\pm.05}$ |

Gini-based method, it consistently achieves the highest accuracy across all datasets evaluated. For this reason, we adopt $\tau$ as the standard criterion in TREWA.

Let us now analyze the effect of the components retained after the wavelet transformation. Table 6 compares three choices: retaining only the approximation coefficients $cA$, only the detail coefficients $cD$, or the sum of them $cA + cD$. This analysis allows us to understand whether the pruning performed by TREWA captures the most relevant information for the task.

Table 6: Performance of the evaluated approaches when training ViT-B from scratch with different frequency-retaining methods.

| Component | FPS ↑ | GFLOPs ↓ | Accuracy ↑ | | |
|---|---|---|---|---|---|
| | | | FMNIST | CIFAR10 | Imagenette |
| $cA$ | **1,132** | **4.616** | **90.64**$_{\pm.12}$ | **67.14**$_{\pm.08}$ | **69.48**$_{\pm.10}$ |
| $cD$ | 1,112 | **4.616** | 88.75$_{\pm.10}$ | 64.90$_{\pm.06}$ | 66.98$_{\pm.05}$ |
| $cA + cD$ | 1,116 | **4.616** | 89.22$_{\pm.13}$ | 65.70$_{\pm.11}$ | 67.13$_{\pm.09}$ |

Table 6 shows that retaining only the low-frequency component $cA$, which represents the approximate information, allows for the best performance in terms of accuracy and inference. This choice allows us to remove the noise associated with high-frequency components while preserving the information most relevant to the task. The combination $cA + cD$ also introduces noisier elements, resulting in worse metrics than using the $cA$ component alone. For this reason, we adopt $cA$ as the reference component to be retained in TREWA.

Finally, we analyze the influence of the mother wavelet $\psi$ used in the DWT transformation. Specifically, we compare four families of wavelets commonly used in signal processing (Rashid et al., 2020): the $haar$, the simplest and most discontinuous; the Daubechies in two variants, $db2$ and $db4$, the former characterized by a more local filter and the latter by a more global one; and finally the Symlet $sym2$, designed to be more symmetrical. The goal of this analysis is to evaluate how the choice of wavelet influences the quality of the coefficients generated by the decomposition and, consequently, the effectiveness of pruning. The results obtained are shown in Table 7.

Table 7: Performance of the evaluated approaches when training ViT-B from scratch with different mother wavelets.

| $\psi$ | FPS ↑ | GFLOPs ↓ | Accuracy ↑ | | |
|---|---|---|---|---|---|
| | | | FMNIST | CIFAR10 | Imagenette |
| $haar$ | **1,132** | **4.616** | **90.64**$_{\pm.12}$ | 67.14$_{\pm.08}$ | **69.48**$_{\pm.10}$ |
| $db2$ | 1,101 | 4.710 | 89.77$_{\pm.07}$ | 66.63$_{\pm.09}$ | 69.41$_{\pm.11}$ |
| $db4$ | 1,049 | 4.859 | 90.11$_{\pm.16}$ | **67.84**$_{\pm.08}$ | 69.53$_{\pm.10}$ |
| $sym2$ | 1,121 | 4.710 | 90.45$_{\pm.09}$ | 66.25$_{\pm.10}$ | 67.17$_{\pm.12}$ |

As shown in Table 7, the $haar$ wavelet is the most efficient in terms of speed and computational cost, while maintaining excellent accuracy performance across all datasets. For this reason, we adopt $haar$ as the reference wavelet $\psi$ in TREWA.

## 6 DISCUSSION

The experimental results, conducted on different datasets and architectures, both in pre-trained and training from scratch settings, show that TREWA is an excellent compromise between accuracy and computational efficiency, as it allows for a significant reduction in GFLOPs and a significant increase in FPS while maintaining the same accuracy, or even improving it in some cases. Compared to established methods such as PM, TREWA is completely free of additional parameters, does not modify the model structure, and can be easily integrated into pre-trained models. This makes it a highly plug-and-play solution, easily adaptable, and suitable for different contexts. Another significant advantage lies in the completely deterministic and interpretable nature of the method: the entire pruning process is guided by an explicit policy based on understandable measures, such as attention entropy, and does not require any learning or optimization phase. Furthermore, unlike many pruning techniques, our strategy is based on selecting only the low-frequency component obtained through DWT, a continuous and differentiable transformation. This is crucial in the context of deep learning, as it allows TREWA to be integrated into the end-to-end training process of the model.

However, TREWA also has some limitations. Currently, it can only be applied to ViT models that use the [CLS] token, as the pruning strategy relies on its attention patterns. Furthermore, the hierarchical nature of DWT only allows reductions in powers of two, limiting the granularity with which the number of tokens retained can be controlled and reducing flexibility in scenarios that require finer-grained adjustment of the computational budget. A further limitation concerns the adaptive threshold based on entropy. When calculating the mean and standard deviation of the values in a batch, TREWA requires batches with at least two samples. With batches of size one, the standard deviation is undefined, and the pruning rule cannot be applied correctly.

## 7 CONCLUSION

In this paper, we present a new token pruning approach for ViTs based on Discrete Wavelet Decomposition. The basic idea is to treat the sequence of tokens input to a Transformer layer as a multidimensional signal, to which the wavelet transform is applied to efficiently separate low-frequency components, which convey stable, long-term semantic information, from high-frequency components, often associated with local details or noise. Starting from the entropy of the [CLS] token attention distribution, we adopt an adaptive strategy that adjusts the pruning level based on the concentration of information in the tokens. In this way, simpler images or those with more focused attention undergo more aggressive pruning, while those with more dispersed attention retain a richer representation. Across FMNIST, CIFAR10, Imagenette, and ImageNet-1k, our approach removes 50–75% of the tokens forwarded to subsequent attention layers, reducing computational load yet preserving the model performance. Because the procedure is deterministic and introduces no additional parameters, it can be integrated into any ViT that uses the classification token, whether pre-trained or trained from scratch, providing an effective way to reduce computational cost. Looking ahead, we plan to explore pruning thresholds based on signal strength, extend the framework to natural language Transformers, and apply it to multimodal models, adapting pruning to the specific characteristics of each modality.

## USE OF LLM

We employed large language models solely to assist with language refinement after the scientific content and experimental results had been fully developed by the authors. No part of the conceptual design, methodology, or data analysis relied on LLM output.

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
