# OpenReview forum: "Token Reduction in Vision Transformers via Discrete Wavelet Decomposition"
_ICLR.cc/2026/Conference — ICLR 2026 Conference Withdrawn Submission_

### Official Review · Reviewer_AojG · 2025-10-17

**Soundness:** 3
**Presentation:** 3
**Contribution:** 3
**Rating:** 4
**Confidence:** 3

**Summary:**

The paper addresses the high computational cost of Vision Transformers (ViTs) by proposing a token pruning method called TREWA. The core idea is to treat token embeddings as signals and apply the Discrete Wavelet Transform (DWT) to separate low-frequency (semantic) and high-frequency (noise) components. Pruning is dynamically controlled per image using the entropy of the [CLS] token's attention distribution, with low-entropy images undergoing more aggressive pruning. Experiments on four datasets (e.g., ImageNet-1k, CIFAR10) and two ViT architectures show that TREWA reduces tokens by 50–75% while maintaining competitive accuracy, outperforming state-of-the-art pruning/merging methods in accuracy-efficiency trade-offs.

**Strengths:**

1. ​​TREWA reinterprets token embeddings as signals and uses DWT for frequency-based pruning, distinct from prior importance-based or merging strategies.

2. The method relies on entropy-driven pruning levels and DWT without introducing learnable parameters.
​​
3. The method is clearly written and explained stepwise, with clear formulations for entropy thresholds and DWT operations.

**Weaknesses:**

1. Incomplete baseline comparisons​​: more recent methods like BAT[1], Token Transforming[2] and EvolutionViT[3] should be discussed and quantitatively compared.

2. The datasets are relatively small and few; more types of task like semantic segmentation, object detection are encouraged to demonstrate the effectiveness of the proposed method.

3. The claim that low-frequency components preserve semantics lacks analytical support beyond empirical results. I would like to see more theoretical analysis but no discussion is shown about this in the paper.

4. The success of the method mainly relies on the Discrete Wavelet Transform (DWT) to separate low- and high-frequency components. Therefore, DWT pruning may degrade performance on datasets such as texture-heavy images that rely heavily on high-frequency details. I would like to see more about failure mode analysis to validate its effectiveness.

5. Limited scalability evidence​​: no results for larger models or high-resolution images.

**Questions:**

1. Marginal improvement: the actual acceleration in tab2 and tab4 is not impressive. Are there any explanation for unsatisfactory acceleration with the least flops? What is the end-to-end latency impact of DWT computations despite the seemingly least computation overhead?

2.  Would adaptive wavelet selection (beyond haar) per dataset further optimize performance?

3.  Are there failure cases (e.g., medical images with critical high-frequency details) where TREWA underperforms?

4. Can the code be released  with DWT implementation details and layer-wise pruning thresholds? This is really important for reproducibility.

5. Would feature importance analysis (e.g., saliency maps) strengthen the claim that low-frequency components retain semantic information?

[1] Beyond Attentive Tokens: Incorporating Token Importance and Diversity for Efficient Vision Transformers.

[2] Token Transforming: A Unified and Training-Free Token Compression Framework for Vision Transformer Acceleration.

[3] EvolutionViT: Multi-objective evolutionary vision transformer pruning under resource constraints.

---

### Official Review · Reviewer_86dk · 2025-10-21

**Soundness:** 3
**Presentation:** 1
**Contribution:** 2
**Rating:** 4
**Confidence:** 4

**Summary:**

This paper proposes a parameter-free token pruning method for Vision Transformers. The key idea is to use discrete wavelet tranform to measure the informativeness of the input. It decomposes them into low- and high-frequency components. Only the low-frequency coefficients—assumed to carry core semantic structure—are retained.  And adpatively filter out the high-frequent components of less informative inputs.

However, there are many pruning methods based on [CLS] token and training free, but the author's discussion is insufficient. It is necessary to emphasize the innovation and unique advantages of this method.

**Strengths:**

The use of DWT for frequency-based token pruning is techically sound and experiments show its effectiveness.

The method is parameter-free, requires no model modification, and can be plug-and-play with any [CLS]-based ViT—even pre-trained ones. This makes it highly deployable.

**Weaknesses:**

1. The writting of methodology part is hard to follow. The text description for the method is confusing especially line 173-177. I recomend the author use equation or figure to present the operations.
2. The threshold for entropy-based policy is computed within a batch. But it`s ineffective for a small batch size.
3. The prunning method for image recongition task has been wildly explored. Maybe the authors should try new senerios for more solid contribution.

**Questions:**

1. As pruning strength is determined by the proposed entropy strategy, how to achieve parallel inference for different pruning strengths in a batch?
2. The reliance on [CLS] attention entropy excludes modern ViT variants that don’t use a [CLS] token (e.g., ViT with global average pooling).
3. However, there are many pruning methods based on [CLS] token and training free, but the author's discussion is insufficient. It is necessary to emphasize the innovation and unique advantages of this method.

---

### Official Review · Reviewer_D5Pw · 2025-10-28

**Soundness:** 2
**Presentation:** 2
**Contribution:** 2
**Rating:** 2
**Confidence:** 4

**Summary:**

This paper aims to reduce the computational cost of Vision Transformers (ViTs) by introducing an adaptive entropy-based pruning policy that estimates image complexity from the attention entropy of the CLS token. Images with low entropy, indicating concentrated attention and low complexity, are pruned more aggressively, while those with high entropy receive lighter pruning. In addition, the method applies the Discrete Wavelet Transform (DWT) to treat token embeddings as multidimensional signals, retaining only the low-frequency components while discarding high-frequency components. This allows the model to effectively eliminate redundant patch tokens and achieve efficient token reduction across layers.

**Strengths:**

Simple and interpretable design: The method is easy to understand and implement, leveraging the DWT to reduce redundant tokens in a mathematically interpretable way. Parameter-free and training-free: It introduces no additional trainable parameters and can be directly applied to pre-trained Vision Transformers without retraining. Consistent efficiency improvement: Experiments show reductions in FLOPs with minimal accuracy degradation, demonstrating a practical trade-off between efficiency and performance.

**Weaknesses:**

Limited novelty and theoretical justification:
Both key components — the use of Discrete Wavelet Transform (DWT) for token reduction and the attention entropy–based pruning policy — are conceptually simple and lack solid theoretical grounding.
The assumption that ViT token embeddings can be treated as spatially continuous signals suitable for frequency decomposition is not well justified, and the entropy-based adaptive rule remains a heuristic without deeper analytical support or empirical sensitivity validation.

Limited experimental scope and applicability:
The experiments are confined to standard Vision Transformer (ViT) models trained on ImageNet-1K, without evaluation on more recent or multimodal architectures. The method has not been applied to modern Vision–Language Models (VLMs), and the paper provides no discussion on its potential applicability or limitations in such settings.

Unstable threshold definition and unclear robustness:
The threshold design is purely statistical and highly sensitive to batch size and data distribution. It remains unclear how the method behaves in extreme cases where all images within a batch are either highly important or uninformative. Moreover, the approach assumes a multi-batch setting; in edge or on-device environments, where single-batch inference is typical, it is uncertain how the adaptive threshold can be applied or adjusted. These limitations raise concerns about the robustness and generalizability of the proposed pruning policy.

Inconsistent efficiency results:
In some experiments, the FPS improvement does not correlate with the reported GFLOPs reduction, suggesting potential computational overhead that is not analyzed or discussed. Specifically, Table 2 and Table 4 show that even when the proposed method achieves the largest GFLOPs reduction, the corresponding FPS improvement remains marginal. Therefore, there is likely a non-negligible computational overhead in the proposed approach that should be explicitly analyzed and discussed.

**Questions:**

1. The paper applies the Discrete Wavelet Transform (DWT) directly to ViT token embeddings. Could the authors clarify the theoretical justification for treating token embeddings as spatially continuous signals suitable for frequency decomposition?

2. The entropy-based adaptive pruning policy appears heuristic. Have the authors conducted any sensitivity or stability analysis to validate how the pruning behavior changes under different entropy distributions or batch sizes?

3. The proposed adaptive threshold relies on batch-level statistics. How would the method behave in extreme cases where all images in a batch have similar entropy values, or when inference is performed on a single sample?

4. The experiments are limited to ViT-B/S models on ImageNet-1K. The proposed method should be evaluated on more recent or multimodal architectures (e.g., CLIP, BLIP, or LLaVA) to demonstrate broader applicability and verify its effectiveness beyond standard ViTs.

5. In Table 2 and Table 4, the method shows the largest GFLOPs reduction but only marginal FPS improvement. Could the authors discuss potential computational overhead or implementation bottlenecks that might explain this discrepancy?

6. Would it be possible to provide visualizations of the pruned and retained tokens to illustrate how the proposed method selects or discards information across layers? Such qualitative results would help clarify whether the pruning strategy effectively preserves semantically important regions.

---

### Note · Authors · 2025-12-16

I have read and agree with the venue's withdrawal policy on behalf of myself and my co-authors.